# Green IoT and Edge AI as Key Technological Enablers for a Sustainable Digital Transition towards a Smart Circular Economy: An Industry 5.0 Use Case

**DOI:** 10.3390/s21175745

**Published:** 2021-08-26

**Authors:** Paula Fraga-Lamas, Sérgio Ivan Lopes, Tiago M. Fernández-Caramés

**Affiliations:** 1Department of Computer Engineering, Faculty of Computer Science, Universidade da Coruña, 15071 A Coruña, Spain; tiago.fernandez@udc.es; 2Centro de Investigación CITIC, Universidade da Coruña, 15071 A Coruña, Spain; 3ADiT-Lab, Instituto Politécnico de Viana do Castelo, Rua da Escola Industrial e Comercial de Nun’Alvares, 4900-347 Viana do Castelo, Portugal; sil@estg.ipvc.pt; 4IT—Instituto de Telecomunicações, Campus Universitário de Santiago, 3810-193 Aveiro, Portugal

**Keywords:** Green IoT, IIoT, edge computing, AI, edge AI, sustainability, digital transition, digital circular economy, Industry 5.0

## Abstract

Internet of Things (IoT) can help to pave the way to the circular economy and to a more sustainable world by enabling the digitalization of many operations and processes, such as water distribution, preventive maintenance, or smart manufacturing. Paradoxically, IoT technologies and paradigms such as edge computing, although they have a huge potential for the digital transition towards sustainability, they are not yet contributing to the sustainable development of the IoT sector itself. In fact, such a sector has a significant carbon footprint due to the use of scarce raw materials and its energy consumption in manufacturing, operating, and recycling processes. To tackle these issues, the Green IoT (G-IoT) paradigm has emerged as a research area to reduce such carbon footprint; however, its sustainable vision collides directly with the advent of Edge Artificial Intelligence (Edge AI), which imposes the consumption of additional energy. This article deals with this problem by exploring the different aspects that impact the design and development of Edge-AI G-IoT systems. Moreover, it presents a practical Industry 5.0 use case that illustrates the different concepts analyzed throughout the article. Specifically, the proposed scenario consists in an Industry 5.0 smart workshop that looks for improving operator safety and operation tracking. Such an application case makes use of a mist computing architecture composed of AI-enabled IoT nodes. After describing the application case, it is evaluated its energy consumption and it is analyzed the impact on the carbon footprint that it may have on different countries. Overall, this article provides guidelines that will help future developers to face the challenges that will arise when creating the next generation of Edge-AI G-IoT systems.

## 1. Introduction

The current digital transformation offers substantial opportunities to industry for building competitive and innovative business models and complex circular supply chains; however, such a transformation also implies severe implications concerning sustainability, since the Information and Communications Technology (ICT) industry has a significant environmental footprint. In order to reach the milestones defined by the United Nations Agenda for Sustainable Development [1] and to implement the visions of circular economy, it is necessary to provide solutions in an efficient and sustainable way during their whole life cycle. Such a sustainable digital transition towards a smart circular economy is enabled by three key technologies: IoT, edge computing, and Artificial Intelligence (AI).

It is estimated that Internet of Things (IoT) and Industrial IoT (IIoT) technologies, which enable ubiquitous connectivity between physical devices, can add, only in industrial applications, USD 14 trillion of economic value to the global economy by 2030 [2]. In addition, the development of the classic view of the Internet of People (IoP) [3] and the Internet Protocol (IP) led to a convergence of IoT technologies over the last two decades, which paved the way for the so-called Internet of Everything (IoE) [4]. Such a concept is rooted in the union of people, things, processes, and data to enrich people’s lives.

The explosion of IoT/IIoT technologies and their potential to pave the way to a more sustainable world (in terms of full control of the entire life cycle of products), can also lead to some pitfalls that represent a major risk in achieving the milestones defined by the UN Agenda for Sustainable Development [1]. As part of the IoT Guidelines for Sustainability that were addressed in 2018 by the World Economic Forum, a recommendation to adopt a framework based on the UN Sustainable Development Goals (SDGs) [1] to evaluate the potential impact and measure the results of the adoption of such recommendations was put forward [2]; however, in 2010–2019, and considering *Goal 12: Ensure sustainable consumption and production* [1], electronic waste grew by 38% and less than 20% has been recycled. Paradoxically, although these technologies have a huge potential for the digital transformation towards sustainability, they are not yet contributing to the sustainable development of the ICT sector. Specifically, such a contribution is expected for the IoT sector, which has been seen as the driving force for a sustainable digital transition. The need for policies that effectively promote the sustainable development of new products and services is crucial and can be seen as a societal challenge in the years to come.

The concept of Green IoT (G-IoT) [5,6] is defined in [7] as: “*energy-efficient procedures (hardware or software) adopted by IoT technologies either to facilitate the reduction in the greenhouse effect of existing applications and services or to reduce the impact of the greenhouse effect of the IoT ecosystem itself*”. In the former case, the use of IoT technologies may help to reduce the greenhouse effect, whereas the latter focuses on the optimization of IoT greenhouse footprints. Moreover, the entire life cycle of a G-IoT system should focus on green design, green production, green utilization, and finally, green disposal/recycling, to have a neutral or very small impact on the environment [7].

IoT devices have increasingly higher computational power, are more affordable and more energy-efficient, which helps to sustain the progress of Moore’s law to bring a sustainable IoT revolution in the global economy [8]; however, this vision directly collides with the advent of the concept of Edge Intelligence (EI) or Edge Artificial Intelligence (Edge-AI), where the processing of the IoT collected data is performed at the edge of the network, which imposes additional challenges in terms of latency, cybersecurity, and more specially, energy efficiency.

This article summarizes the most relevant emerging trends and research priorities for the development of Edge-AI G-IoT systems in the context of sustainability and circular economy. In particular, the following are the main contributions of the article:The essential concepts and background knowledge necessary for the development of Edge-AI G-IoT systems are detailed.The most recent Edge-AI G-IoT communications architectures are described together with their main subsystems to allow future researchers to design their own systems.The latest trends on the convergence of AI and edge computing are detailed. Moreover, a cross-analysis is provided in order to determine the main issues that arise when combining G-IoT and Edge-AI.The energy consumption of a practical Industry 5.0 application case is analyzed to illustrate the theoretical concepts introduced in the article.The most relevant future challenges for the successful development of Edge-AI G-IoT systems are outlined to provide a roadmap for future researchers.

The remainder of this article is structured as follows. Section 2 introduces the essential concepts that will be used in the article. Section 3 analyzes the main aspects related to the development of G-IoT systems, including their communications architecture and their main subsystems. Section 4 analyzes the convergence of AI and edge computing to create Edge-AI systems. Section 5 provides a cross-analysis to determine the key issues that arise when combining G-IoT and Edge-AI systems. Section 6 presents a practical Industry 5.0 application case and evaluates the energy consumption of a mist computing Edge-AI G-IoT model. Section 7 outlines the main future challenges that stand in the way of leveraging Edge-AI G-IoT systems. Finally, Section 8 is devoted to the conclusions.

## 2. Background

### 2.1. Digital Circular Economy

#### 2.1.1. Circular Economy

Circular Economy (CE) promotes an enhanced socio-economic paradigm for sustainable development. It aims to fulfill current needs without jeopardizing the needs of future generations under three dimensions: economic, social, and environmental. The European Green Deal [9], Europe’s new agenda for sustainable growth, is an ambitious action plan to move to a clean circular economy, to restore biodiversity, to reduce emissions by at least 55% by 2030, and to become the world’s first climate neutral continent by 2050. The EC strategy is well aligned with the United Nations (UN) 2030 Agenda for Sustainable Development [10]. The 17 Sustainable Development Goals (SDGs) are at the heart of the EU policymaking across all sectors.

CE reforms current linear “take-make-dispose” economic models based on unsustainable mass production and consumption and proposes a new model that is restorative by design (materials, components, platforms, resources, and products add as much value as possible throughout their life cycle). Such a model also aligns the needs of the different stakeholders through business models, government policies, and consumer preferences [11]. At the end of their lifetime, much of these products and components are regenerated and/or recycled.

The European Commission adopted a new Circular Economy Action Plan (CEAP) in March 2020, as one of the main key elements of the European Green Deal [12]. Such an action plan promotes initiatives along the entire life cycle of products, from design to the end of their lifetime, encouraging sustainable consumption and waste reduction. According to the World Economic Forum [13], achieving a CE transition will require unprecedented collaboration, given that, in 2019, only 8.6% of the world was circular, although CE can yield up to USD 4.5 trillion in economic benefits in 2030 [14].

#### 2.1.2. Digital Circular Economy (DCE)

Data centers and digital infrastructures require substantial levels of energy. ICT accounts for 5 to 9% of the total electricity demand with a potential increase to 20% by 2030 [15]. In addition, materials (e.g., physical resources, raw materials) linked to the digital transformation are also a problem: the world produces over 50 million tons of electronic and electrical waste (e-waste) annually and just 20% is formally recycled. Such an amount of waste will reach 120 million tons annually by 2050 [16].

The challenge posed by the increase in digital technologies requires the application of circular economy principles to the digital infrastructure. While currently, the focus of the sector is mainly on meeting the needs in a sustainable way (e.g., energy efficiency and cybersecurity), the supply of critical raw materials will be an issue in the coming years. Moreover, the opportunities provided by the DCE to the digital transition should be also explored (e.g., new business models, new markets, and reduced information asymmetry).

#### 2.1.3. G-IoT and Edge-AI for Digital Circular Economy (DCE)

Digital technologies are a key enabler for the upscaling of the circular economy, as they allow for creating and processing data required for new business models and complex circular supply chains. In addition, they can close the information and transparency gaps that currently slow down the scale-up of DCE.

There is a need for further integration of digital enabling technologies such as functional electronics (e.g., nanoelectronics, flexible, organic and printed electronics or electronic smart systems), blockchain [17], edge computing [18], UAVs [19], 5G/6G [20], big data, and AI [21] into existing circular business approaches to provide information and additional services.

Specifically, G-IoT and Edge-AI have the potential to substantially leverage the adoption of DCE concepts by organizations and society in general in two main ways. First, by considering an open G-IoT architecture [11], where G-IoT devices have circularity enabling features (e.g., end-to-end cybersecurity, privacy, interoperability, energy harvesting capabilities). Second, by having a network of Edge-AI G-IoT connected devices that provide fast smart services and real-time valuable information to the different stakeholders (e.g., designers, end users, suppliers, manufacturers, and investors). Thus, supply chain visibility and transparency of the product, of the production system, and the whole business, are ensured. Moreover, stakeholders can rely on real-time accurate information to make the right decisions at the right time to use resources effectively, to improve the efficiency of the processes, and to reduce waste. Furthermore, asset monitoring and predictive maintenance can increase product lifetime. Figure 1 illustrates the previous concepts and provides an overall view of the main areas impacted by the combined use of G-IoT and Edge-AI.

### 2.2. Industry 5.0 and Society 5.0

The Industry 5.0 paradigm is still being characterized by industry and academia, but the European Commission has already defined its foundations, due to the impact that such a concept will have in the coming years for the European industry [22]. The proposed concept seeks to correct some Industry 4.0 aspects that have not been properly addressed or that have become controversial due to forgetting essential values such as social fairness and sustainability. Thus, according to the European Commission, the foundations of Industry 5.0 have to be completely aligned with societal goals and to aim higher than just considering jobs and economic growth. As a consequence, Industry 5.0 is focused on sustainable manufacturing and industrial operator well-being [23].

It is important to note that Industry 5.0 has not been conceived as a complete industrial revolution, but as a complement to Industry 4.0 that contemplates aspects that link emerging societal trends to industrial development [24]; therefore, the Industry 5.0 paradigm looks for the improvement of smart factory efficiency through technology, while minimizing environmental and social impacts.

It is also worth pointing out that the vision of Industry 5.0 according to the European Commission seems to be clearly inspired by a previous concept: Society 5.0. Such a concept was first put forward by the Japanese government in 2015 [25] and later (in 2016) it was fostered by Keidanren, one of the most relevant business federations of Japan [26]. Society 5.0 goes beyond industrial company digitalization and proposes a collaborative strategy for the whole Japanese society, as it happened throughout history with the four previous society revolutions: Society 1.0 and Society 2.0 are related to hunters and gatherers; Society 3.0 is associated with the industrial revolution that occurred at the end of the 18th century; and Society 4.0 arose from the information-based economies related to the spread of the Internet and on industrial digitalization. As a continuation to Society 4.0, Society 5.0 still looks for expanding economic development, but, at the same time, it keeps in mind societal and environmental concerns.

### 2.3. Technology Enablers

In order to reach the UN Sustainable Development Goals and to implement the visions of the digital circular economy, Society 5.0, and Industry 5.0, it is necessary to provide solutions to integrate the physical and virtual worlds in an efficient and sustainable way. Thus, the next subsections describe the three key technology enablers that this article is focused on and that need to be optimized to make our daily lives and industrial processes greener.

#### 2.3.1. IoT and IIoT

The term IoT refers to a network of physical devices (i.e., “things”) that can be connected among themselves and with other services that are deployed over the Internet. Such devices are usually composed of sensors, actuators, communications transceivers, and computationally constrained processing units (e.g., microcontrollers). IoT devices have multiple applications in fields such as appliance remote monitoring [27], home automation [28], or precision agriculture [29]. The adaptation of the IoT principles to industrial environments is referred to as IIoT and allows for deploying many remotely monitored and controlled sensors, actuators, and smart machinery in industrial scenarios [30,31,32].

#### 2.3.2. Cloud and Edge Computing

Most current IoT applications are already deployed on cloud computing based systems since they allow for centralizing data storage, processing, and remote monitoring/interaction; however, such centralized solutions have certain limitations. The cloud itself is considered a common point of failure, since attacks, vulnerabilities, or maintenance tasks can block it and, as a consequence, the whole system may stop working [33]. In addition, it is important to note that the number of connected IoT devices is expected to increase in the next years [34] and, consequently, the number of predicted communications with the cloud may overload it if it is not scaled properly.

Due to the previous constraints, in recent years, new architectures have been proposed. In the case of edge computing, it is aimed at offloading the cloud from tasks that can be performed by devices placed at the edge of an IoT network, close to the end IoT nodes. Thus, different variants of the edge computing paradigm have been put forward, such as fog computing [35], proposed by Cisco to make use of low-power devices on the edge, or cloudlets [36], which consist of high-end computers that perform heavy processing tasks on the edge [37,38].

#### 2.3.3. AI

AI is a field that looks for adding intelligence to machines [39]. Such intelligence can be demonstrated in the form of recommendation systems, human-speech recognition solutions, or autonomous vehicles that are able to make decisions on their own. The mentioned examples are able to collect information from the real world and then process it in order to provide an output (i.e., a solution to a problem). In some cases, AI systems need to learn previously how to solve a specific problem, so they need to be trained.

In the case of IoT systems, AI systems receive data from the deployed IoT nodes, which usually collect them from their sensors. In traditional IoT architectures, such data are transmitted to a remote cloud where they are processed by the AI system and a result is generated, which usually involves making a decision that is communicated to the user or to certain devices of the IoT network.

The problem is that real-time IoT systems frequently cannot rely on cloud-based architectures, since latency prevents the system from responding timely. In such cases, the use of Edge-AI provides a solution: edge computing devices are deployed near the IoT end nodes, so lag can be decreased, and IoT node requests are offloaded from the cloud, thus avoiding potential communications bottlenecks when scaling the system.

Although Edge-AI is a really useful technology for IoT systems, their combination derives into systems that can consume a significant amount of energy, so Edge-AI IoT systems need to be optimized in terms of power consumption. The next sections deal with such a problem: first, the factors that impact the development of energy-efficient (i.e., green) IoT systems are studied and then the power consumption of Edge-AI systems is analyzed.

## 3. Energy Efficiency for IoT: Developing Green IoT Systems

### 3.1. Communications Architectures for G-IoT Systems

Before analyzing how G-IoT systems try to minimize energy consumption, it is first necessary to understand which components make up an IoT architecture. Thus, Figure 2 depicts a cloud-based architecture, currently the most popular IoT architecture, which is built around the cloud computing paradigm. Such a cloud collects data from remote IoT sensors and can send commands to IoT actuators. The cloud is also capable of interacting with third-party services (usually hosted in servers or other cloud computing systems) and with remote users, to whom it provides management software.

Cloud-based IoT systems have allowed the spread of IoT systems, but, since they are commonly centralized, they suffer from known bottlenecks (e.g., Denial of Service (DoS) attacks) and from relatively long response latency [33]. To tackle such issues, in recent years, new IoT paradigms have been explored, such as edge, fog, or mist computing [35,40], which offload the cloud from certain tasks to decrease the amount of node requests and to reduce latency response. In the case of edge computing, it adds a new layer between the cloud and the IoT devices (where the gateway is placed in Figure 2) to provide them with fast-response services through edge devices such as cloudlets or fog computing gateways [41]. Fog computing gateways are computationally constrained devices (e.g., routers and Single-Board Computers (SBCs)) that provide support for physically distributed, low-latency, and Quality of Service (QoS) aware applications [35,37]. Cloudlets allow for providing real-time rendering or compute-intensive services, which require deploying high-end PCs in the local network [36]. Regarding mist computing devices, they perform tasks locally at the IoT nodes and can collaborate with other IoT nodes to perform complex tasks without relying on a remote cloud [37,40,42,43,44]. Thus, mist nodes reduce the need for exchanging data to the higher layers of the architecture (thus saving battery power), but, in exchange, they are responsible for carrying out multiple tasks locally.

Figure 3 depicts an example of mist computing based architecture. In this figure, for the sake of clarity, no edge computing layer is included, but it is standard to make use of it in practical applications [42]. The two layers that are present are the cloud layer, which works similarly to the previously described architectures, and the mist computing layer, which is composed by mist nodes. Such nodes embed additional hardware to perform the necessary local processing tasks. In addition, it is worth noting that mist nodes often can communicate directly among themselves, thus avoiding the need for using intermediate gateways.

After analyzing the previous architectures, it can be stated that, to create G-IoT systems, it is necessary to consider the efficiency of the hardware and software of their main components: the IoT nodes, the edge computing devices, and the cloud. The next sections delve into such a topic, reviewing the most relevant contributions of the state of the art.

### 3.2. Types of G-IoT Devices

The development and deployment of efficient G-IoT devices is conditioned by their hardware and software. It is also important to note that the requirements of the G-IoT devices differ significantly: G-IoT nodes do not have the same energy consumption needs as edge devices (e.g., fog computing gateways, cloudlets, Mobile Edge Computing (MEC) hardware) or the cloud. Nonetheless, all the involved hardware have in common the fact that it is essential to select the main parts that allow for optimizing their energy efficiency (the control and power subsystems), and the communications interfaces. Regarding software, the control software, the implemented communications protocols, and the used security algorithms are essential when minimizing energy consumption. The next subsections analyze such hardware and software components in order to guide future G-IoT developers.

### 3.3. Hardware of the Control and Power Subsystems

There are different approaches to maximize the energy efficiency of IoT deployments. One of the most important is to find the right trade-off between the different capabilities of the control hardware and their energy consumption. Currently, the most popular IoT nodes are based on microcontrollers. Such devices are usually cheap, have enough processing power to perform control tasks, can be easily reprogrammed, and have low-energy consumption. There are other more sophisticated alternatives, such as Digital Signal Processors (DSPs), System-On-Chips (SOCs), Central Processing Units (CPUs), Field-Programmable Gate Arrays (FPGAs), Complex Programmable Logic Devices (CPLDs), Graphics Processing Units (GPUs), and Application-Specific Integrated Circuits (ASICs).

DSPs are usually power efficient, especially certain models designed specifically for low power consumption (e.g., Texas Instruments TMS320C5000). Central Processing Units (CPUs) (e.g., Intel Xeon) are general-purpose processing units that offer an adequate trade-off between performance and power consumption, but they are usually optimized for high-speed and parallel processing. With respect to SoCs, they integrate medium-to-high performance microcontrollers and peripherals, so they consume more power than traditional microcontrollers, but they are more appropriate for lightweight systems. In the case of FPGAs, they offer very good performance for executing deterministic tasks, but its programming is not as easy as with microcontrollers, and they require to power the used logic continuously. There are also hybrid solutions that combine the benefits of FPGAs and CPUs, known as Field-Programmable Systems-on-Chips (FPSoCs) [45]. In the case of CPLDs, they can execute tasks faster than FPGAs, but their maximum allowed design complexity is inferior to the one offered by FPGAs. GPUs were created to offload graphic computation from the CPUs, but current products can include several thousands of cores designed for the efficient execution of complex functions. Regarding ASICs, they offer even higher performance than FPGAs and other embedded devices, since they are optimized for power consumption, but their development cost is very high (usually in the order of millions of dollars).

Besides choosing the right control hardware, it is necessary to optimize the power subsystems. Most current IoT node deployments rely on batteries. Such batteries can store a finite amount of energy, and they need to be replaced or recharged frequently. Maintenance tasks are costly and cumbersome, especially in large deployments, industrial confined spaces, or remote areas. In addition, such tasks are critical when developing power-hungry applications. Battery replacement also leads to a heavy carbon footprint due to the use of scarce raw materials, the battery manufacturing process, and the involved recycling processes; therefore, there is a need for self-sustainable solutions such as environmental energy harvesting. Such solutions exploit ubiquitous energy sources in the deployment area without requiring external power sources and ease maintenance tasks. The most common harvesting techniques are related to solar and kinetic energy sources. Examples of different energy harvesting techniques are presented in [46,47,48,49].

### 3.4. Communications Subsystem

G-IoT devices can make use of different technologies for their communications interfaces. The communications with the cloud are usually through the Internet or a wired intranet, so this section focuses on the energy efficiency of the wireless communications technologies used by G-IoT nodes and edge devices. Table 1 compares the characteristics of some of the most relevant communications technologies according to their power consumption, operating band, maximum range, expected data rate, their relevant features, and main applications.

G-IoT node communications need to provide a trade-off between features and energy consumption. For example, Near-field Communication (NFC) [50] is able to deliver a reading distance of up to 30 cm, but NFC tags usually do not need to make use of batteries since they are powered by the readers through inductive coupling. NFC is a technology derived from Radio Frequency Identification (RFID), which, despite certain security constraints [51], in recent years, has experienced significant growth in home and industrial scenarios [52,53] thanks to its very low power consumption. It must be noted that RFID and NFC are essentially aimed at identifying items, but they can be used for performing regular wireless communications among G-IoT nodes (e.g., for reading embedded sensors). Nonetheless, there are technologies that have been devised to provide more complex interactions. For instance, Bluetooth implementations such as Bluetooth Low Energy (BLE) can provide wireless communications distances between 10 and 100 m [54] and very low energy consumption thanks to the use of beacons [55], which are a sort of lightweight IoT devices able to transmit packets at periodic time intervals.

The widely popular Wi-Fi (i.e., IEEE 802.11 standards) can also provide indoor and outdoor coverage easily and inexpensively for IoT nodes; however, its energy consumption is usually relatively high and proportional to the speed rate. Nonetheless, new IEEE 802.11 standards have been proposed in recent years so as to reduce energy consumption. For instance, Wi-Fi Hallow offers low power consumption (comparable with Bluetooth) while maintaining high data rates, and a wider coverage range.

In terms of green communications, the following are currently the most popular and promising technologies:ZigBee [56]. It was conceived for deploying Wireless Sensor Networks (WSNs) that are able to provide overall low energy consumption by being asleep most of the time, just waking up periodically. In addition, it is easy to scale ZigBee networks, since they can create mesh networks to extend the IoT node communications range.LoRA (Long-Range Wide Area Network) and LoRAWAN [57]. These technologies have been devised to deploy Wide Area IoT networks while providing low energy consumption.Ultrawideband (UWB). It is able to provide low-energy wide-bandwidth communications as well as centimeter-level positioning accuracy in short-range indoor applications. Mazhar et al. [58] evaluate different UWB positioning methods, algorithms, and implementations. The authors conclude that some techniques (e.g., hybrid techniques combining both Time-of-Arrival (TOA) and Angle-of-Arrival (AOA)), although more complex, are able to offer additional advantages in terms of power consumption and performance.Wi-Fi Hallow/IEEE 802.11ah. In contrast to Wi-Fi, it offers very low energy consumption by adopting novel power-saving strategies to ensure an efficient use of energy resources available in IoT nodes. It was specifically created to address the needs of Machine-to-Machine (M2M) communications based on many devices (e.g., hundreds or thousands), long range, sporadic traffic needs, and substantial energy constraints [59].

### 3.5. Green Control Software

There is a significant number of recent publications that propose different techniques and protocols for network control and power saving. For instance, there are G-IoT protocols for interference reduction, optimized scheduling (e.g., switching selectively inactive sensor nodes and put them into deep sleep mode), resource allocation and access control, temporal and spatial redundancy, cooperative techniques in the network, dynamic transmission power adjustment, or energy harvesting [6].

Power-efficient network routing is also a hot topic. For instance, Xie et al. [60] reviewed recent works on energy-efficient routing and propose a novel method for relay node placement. Other authors focused on solutions for service-aware clustering [61]. Another interesting work can be found in [62], where the authors present an energy-efficient IoT architecture able to predict the adequate sleep interval of sensors. The experimental results show significant energy savings for sensor nodes and improved resource utilization of cloud resources. Nonetheless, this solution is not valid for applications with real-time requirements or that require constant availability. Finally, recent approaches such as [63] proposed solutions that combine distributed energy harvesting-enabled mobile edge computing offloading systems with on-demand computing resource allocation and battery energy level management.

### 3.6. Energy Efficient Security Mechanisms

A number of attacks can be performed to break the confidentiality, integrity, and availability of IoT/IIoT networks (e.g., jamming, malicious code injection, Denial of Service (DoS) attacks, Man-in-the-Middle (MitM) attacks, and side-channel attacks) [64]. In order to have protection for such attacks, secure deployment of G-IoT networks should involve three main elements: architecture, hardware, and the security mechanisms across the different devices.

The resource-constrained nature of IoT devices, specially IoT nodes, imposes limitations on the inclusion of complex protocols to encrypt and secure communications [65]. This is particularly challenging when implementing cryptosystems that require substantial computational resources. Hash functions, symmetric cryptography, and public-key cryptosystems (i.e., asymmetric cryptographic systems such as Rivest–Shamir–Adleman (RSA) [66], Elliptic Curve Cryptography (ECC) [67,68], or Diffie–Hellman (DH) [69]) are among the most used cryptosystems.

Public-key cryptosystems are essential for authenticating transactions and are part of Internet standards such as Transport Layer Security (TLS) (TLS v1.3 [70]), currently the most-extended solution for securing TCP/IP communications. Regarding cipher suites recommended for TLS, Rivest–Shamir–Adleman (RSA) and Elliptic Curve Diffie–Hellman Ephemeral (ECDHE) are the most popular ones.

The execution of cryptographic algorithms must be fast and energy efficient, but still provide adequate security levels. Such a trade-off has attracted scientific attention, which is currently an active area of research [71], especially since recent advances in computation have made it easy to break certain schemes (e.g., 1024-bit RSA is broken [72]); however, there are few articles in the literature that address the impact of security mechanisms on energy consumption for G-IoT systems. For instance, in [42], the authors compare the energy consumption of different cryptographic schemes, showing that, at the same security level, some schemes are clearly more efficient in terms of energy and data throughput than others when executed on certain IoT devices.

Moreover, hardware acceleration can be used for keeping energy consumption and throughput values at a reasonable level when executing public-key cryptography algorithms [73]. Furthermore, the use of specific hardware can also speed up the execution of cryptographic algorithms such as hash algorithms [74] or block ciphers [75].

### 3.7. G-IoT Carbon Footprint

The concept of carbon footprint (or carbon dioxide emissions coefficient) measures the amount of greenhouse gases (including CO_2_) caused by human or non-human activities. In the case of the development and use of a technology, it involves a carbon footprint related to its life cycle: from the design stage to the recycling of products. This is especially critical for IoT, since a large number of connected devices is expected in the coming years (up to 30.9 billion in 2025 [76]), which will consume a significant amount of electricity and, as a consequence, a high volume of carbon dioxide will be emitted into the environment. G-IoT has emerged as an attractive research area whose objective is to study how to minimize the environmental impact related to the deployment of IoT networks in smart homes, factories, or smart cities [77].

The following are some of the challenges that must be faced in order to reduce IoT network carbon footprint and environmental impact [78,79]:Hardware power consumption. The used IoT hardware is the basis for the IoT network, so its energy consumption should be as energy efficient as possible while preserving its functionality and required computing power.IoT node software energy consumption. Software needs to be optimized together with the hardware, so developers need to introduce energy-aware constraints during the development of G-IoT solutions. Such optimizations are especially critical for certain digital signal processing tasks such as compression, feature extraction, or machine learning training [80].IoT protocol energy efficiency. The IoT relies on protocols that enable communicating between the multiple nodes and routing devices involved in an IoT network. As a consequence, such protocols need to be energy efficient in terms of software implementation and should consider the minimization of the usage of communication interfaces. For instance, Peer-to-Peer (P2P) protocols are well-known for being intensive in terms of the number of communications they manage, although some research has been dedicated to reducing their energy consumption [81,82,83,84,85].RF spectrum management optimization. The increasing number of deployed IoT nodes will derive into the congestion of the RF spectrum, so its management will need to be further optimized to minimize node energy consumption [77].Datacenter sustainability. As the demand for IoT devices grows, ever-increasing amounts of energy are needed to power the datacenters where remote cloud services are provided. This issue is especially critical for corporations such as Google or Microsoft, which rely on huge data centers and, in fact, the U.S. Environmental Protection Agency (EPA) already warned about this problem in 2007 [86]. As a consequence of such a warning, carbon footprint estimations were performed in order to determine the emissions related to the construction and operation of a datacenter [87].Data storage energy usage. In cloud-centric architectures, most of the data are stored in a server or in a farm of servers in a remote datacenter, but some of the latest architectures decentralize data storage to prevent single-point-of-failure issues and avoid high operation costs. Thus, for such decentralized architectures, G-IoT requires minimizing node energy consumption and communications. This is not so easy, since devices are physically scattered, and they usually make use of heterogeneous platforms whose energy optimization may differ significantly.Use of green power sources. IoT networks can become greener by making use of renewable power sources from wind, solar, or thermal energy. IoT nodes can also make use of energy-harvesting techniques to minimize their dependence on batteries or extend their battery life [46,47,48,49]. Moreover, battery manufacturing and end-of-life processes have their own carbon footprint and impact the environment with their toxicity. Furthermore, IoT architectures can be in part powered through decentralized green smart grids, which can collaborate among them to distribute the generated energy [78].Green task offloading. Traditional centralized architectures have tended to offload the computing and storage resources of IoT devices to a remote cloud, which requires additional power consumption and network communications that are proportional to the tasks to be performed and to the latency of the network. In contrast, architectures such as the ones described in Section 3.1, can selectively choose which tasks to offload to the cloud. Thus, most of the node requests are processed in the edge of the network, which reduces latency and network resource consumption due to the decrease in the number of involved gateways and routers [88]. Nonetheless, G-IoT designers must be aware of the energy implications of decentralized systems [89].

## 4. AI and Edge Computing Convergence

As previously mentioned in Section 2.3.3, AI can be broadly defined as a science capable of simulating human cognition to incorporate human intelligence into machines. Machine Learning (ML) can be seen as a specific subset of AI, as a technique for training algorithms that focuses on empowering computer systems with the ability to learn from data, perform accurate predictions, and therefore, make decisions. The training stage in ML involves the collection of huge amounts of data (train set) to train an algorithm that allows the machine to learn from the processed information. Then, after training, the algorithm is used for inference in new data [90]. Deep Learning (DL) is a subset of ML that can be seen as the natural evolution of ML. DL algorithms are inspired by the human brain cognitive processing patterns (i.e., by its ability for pattern identification and classification), using DL algorithms that are trained to perform the same tasks in computer systems. By analogy, the human brain typically attempts to interpret a new pattern by labeling it and performing subsequent categorization [91]. Once new information is received, the brain attempts to compare it to a known reference before reasoning, which is conceptually what DL algorithms perform (e.g., Artificial Neural Networks (ANNs) algorithms aim to emulate the way the human brain works). In [91], Samek et al. identified two major differences between ML and DL:DL can automatically identify and select the features that will be used in the classification stage. In contrast, ML requires the features to be provided manually (i.e., unsupervised vs. supervised learning).DL requires high-end hardware and large training data sets to deliver accurate results, as opposed to ML, which can operate in low-end hardware with smaller data sets in the training stage (i.e., ML is typically adopted in resource contained embedded hardware).

The use of such AI techniques is highly dependent, not only on the hardware specifications and the available computational power, but also on the adopted inference approach [92].

### 4.1. AI-Enabled IoT Hardware

AI-enabled IoT devices are paving the way to implement new and increasingly complex cyber–physical systems (CPS) in distinct application domains [93,94,95]. The increasing complexity of such devices is typically specified based on SWaP requirements (i.e., reduced Size, Weight, and Power) [96]. When considering the IoT/IIoT ecosystems, changes in SWaP requirements, and also in unit cost, may impact the overall performance and functionality of the end devices, since the number of devices tends to increase at a steady pace, the cost per unit becomes more and more relevant. Note that the number of devices deployed is expected to increase massively in the coming years, with many of these devices operating as sensors and/or actuators, which will demand increasing processing power enabling effective edge AI deployment. On the other hand, portability is also relevant, and therefore, power will often come from an external battery or an energy harvesting subsystem, which imposes several challenges in the design of AI-enabled IoT devices. For example, in [97], a study regarding low-power ML architectures has been put forward and results have shown that sub-mW power consumption can potentially be deployed in “always-ON” AI-enabled IoT nodes.

#### 4.1.1. Common Edge-AI Device Architectures

The G-IoT hardware previously described in Section 3.2 has evolved in recent years as illustrated in Figure 4 in order to provide AI-enable functionality. Thus, basic IoT hardware (represented at the top of Figure 4), typically uses a traditional computing approach that combines an embedded processor (CPU) or a microcontroller (MCU) with on-board memory, sensor/actuator interfacing—digital (e.g., SPI, I2C, 1-Wire) and analog (ADCs, DACs) inputs/outputs—and basic connectivity (e.g., Wi-Fi, Bluetooth).

AI-enabled IoT device architectures (depicted in the middle of Figure 4), use a near-memory computing approach based on a multicore CPU or FPGA, and typically includes external sensors and actuators, and extended connectivity options such as NB-IoT, LoRaWAN, or 5G/6G support.

Lastly, an AI-specific IoT device also includes cognitive capabilities and typically uses an in-memory computing approach, which may be supported by a dedicated AI SoC, specifically included to execute learning algorithms (this architecture is depicted at the bottom of Figure 4). IoT devices are getting increasingly powerful and computationally efficient as new SoCs with integrated AI chips become available. For example, the usage of FPGAs in AI-enabled IoT devices allows high-speed inference, parallel execution, and the implementation of application-specific computational architectures without the need for expensive ASICs; however, the total power consumption may be a problem when using FPGAs in power-sensitive applications [96].

#### 4.1.2. Embedded AI SoC Architectures

Embedded AI SoCs are used in specific IoT architectures [98], allowing for the execution of ML algorithms directly on the end device, and therefore detecting patterns and trends in data, and enabling the transmission of low-bandwidth data streams with contextual information to enhance decision-making and empower prognosis throughout the use in-device prediction models and ML, as it is represented at the bottom in Figure 4. In [96], Mauro et al. achieved high performance in power saving for both logic and SRAM design, using Binary Neural Networks (BNNs). BNNs enable the deployment of deep models on resource-constrained devices [99], because they may be trained to produce outcomes comparable to full-precision alternatives while maintaining a smaller footprint, a more scalable structure, and better error resilience. Such characteristics enable the implementation of completely programmable SoC IoT end-devices capable of performing hardware-accelerated and software-defined algorithms at ultra-low power, reaching 22.8 Inference/s/mW while using 674 μW [98].

#### 4.1.3. AI-Enabled IoT Hardware Selection Criteria

Running an AI model at an AI-enable IoT device presents four main advantages when compared with the classical cloud-based approach:Reliable Connectivity: data can be gathered and processed on the same device instead of relying on a network connection to transmit data to the cloud, which reduces the probability of network connection problems.Reduced Latency: when processing is performed locally, all communications-related latencies are avoided, resulting in an overall latency that converges to the inference latency.Increased Security and Privacy: reducing the need for communicating between the IoT edge device and the cloud means reducing the risk that data will be compromised, lost, stolen, or leaked.Bandwidth Efficiency: reducing the communications between IoT edge devices and the cloud, also reduces bandwidth needs and the overall communications cost.

Table 2 compiles several AI-enabled IoT hardware boards that are able to run ML libraries, such as Tensorflow Lite [100]. TensorFlow Lite is an open-source ML library specifically designed for resource-constrained IoT devices, that typically use MCU-based architectures.

### 4.2. Edge Intelligence or Edge-AI

Typically, in cloud-centric architectures, IoT devices can transfer data to the cloud using an Internet gateway. In this architecture, the raw data produced by IoT devices are pushed to a centralized server without processing; however, since IoT devices are becoming more efficient and powerful, new possibilities arise at the network edge, enabling real-time intelligent processing with minimal latency. Edge Intelligence (EI) or Edge-AI are the common names given to this approach, and its performance is often expressed in terms of model accuracy and overall latency [107].

A common IoT device (also known as a “dumb” device) tends to generate large quantities of raw and low-quality data, which may have no operational relevance. In most cases, data are noisy, intermittent, or change slowly, being useless in specific periods. Moreover, the management and transmission of these useless data streams consume vital power and tend to be bandwidth-intensive. On the other hand, the inclusion of in-device/edge intelligence results in the reduction in the data dimension by turning data into relevant information, lowering power consumption, latency, and the overall bandwidth needs. Intelligence at the edge of the network enables the distribution of the computational cost among edge devices. In this computational approach, data can be classified and aggregated before its transmission up to the cloud. By using this approach, only information with historical value is archived, which can be later used for tuning prediction models and optimizing the cloud-based processing.

#### 4.2.1. Model Inference Architectures

The three major Edge-AI computing paradigms are [108]:(i)On-device computation: it relies on AI techniques (e.g., Deep Neural Networks (DNNs)) that are executed on the end device.(ii)Edge-based computation: it computes on edge devices the information collected from end devices.(iii)Joint computation: it allows for processing data on the cloud during training and inference stages.

Given the limited resources that are typically available in most IoT devices, bringing AI to the edge can be challenging. Reducing model inference time has been implemented successfully at the cost of decreasing the overall model inference accuracy. According to Merenda et al. [109], to effectively run an AI model (after the compression stage) in an embedded IoT device, the hardware selection must be carefully performed.

#### 4.2.2. Edge-AI Levels

Besides the well-known Cloud Intelligence (CI), which consists in training and inferencing the DNN models fully in the cloud, EI, as described in [110], can be classified into the six levels depicted in Figure 5. The quantity of data sent up to the cloud tends to decrease as the level of EI increases, resulting in lower communications bandwidth and lower transmission delay; however, this comes at the cost of increased computational latency and energy consumption at the network’s edge (including IoT nodes), implying that the EI level is application-dependent and must be carefully chosen based on several criteria: latency, energy efficiency, and privacy and communications bandwidth cost.

Inference and training are the two main computing stages in an NN. Depending on the Edge-AI level (as illustrated in Figure 5), the computational power is typically distributed between the IoT node or the edge layer, which requires increased computational power. In recent years, AI-specific hardware accelerators have enhanced high-performance inference computation at the edge of the network, namely in embedded and resource-constrained devices. For example, in [111], Karras et al. present an FPGA-based SoC architecture to accelerate the execution of ML algorithms at the edge. The system presents a high degree of flexibility and supports the dynamic deployment of ML algorithms, which demonstrate an efficient and competitive performance of the proposed hardware to accelerate AI-based inference at the edge. Another example is presented in [112] by Kim et al., where they propose a co-scheduling method to accelerate the convolution layer operations of CNN inferences at the edge by exploiting parallelism in the CNN output channels. The developed FPGA-based prototype presented a global performance improvement of up to 200%, and an energy reduction between 14.9% and 49.7%. Finally, in [113], the authors introduce NeuroPipe, a hardware management method that enables energy-efficient acceleration of DNNs on edge devices. The system incorporates a dedicated hardware accelerator for neural processing. The proposed method enables the embedded CPU to operate at lower frequencies and voltages, and to execute faster inferences for the same energy consumption. The provided results show a reduction in energy consumption of 11.4% for the same performance.

#### 4.2.3. Embedded ML

Conventional IoT devices are ubiquitous and low-cost, but natively resource-constrained, which limits their usage in ML tasks; however, data generated at the edge are increasingly being used to support applications that run ML models. Until now, edge ML has been predominantly focused on mobile inference, but recently several embedded ML solutions have been developed to operate in ultra-low-power devices, typically characterized by its hard resource constraints [97]. Recently, a new field of ML, known as Tiny ML, was put forward to enable inference at the edge endpoints. ML inference at the edge can optimize the overall computational resource needs, increases privacy within applications, and enhances system responsiveness. TinyML, which has been coined due to its ML inference power consumption of under a milliWatt, overcomes the power limitations of such devices, enabling low-power and low-cost distributed machine intelligence. TinyML is an open-source ML framework specifically designed for resource-constrained embedded devices. It is fully compatible with several low-cost, globally accessible hardware platforms and was designed to streamline the development of embedded ML applications [114].

TinyML technologies and applications target battery-operated devices, including hardware, algorithms, and software for on-device inference and data analytics at the edge. In [115], MLCommons, an open engineering consortium, presented a recent benchmark (MLPerf™ Tiny Inference v0.5). This inference benchmark suite targets ML use cases on embedded devices by measuring how rapidly a trained NN can process new data in ultra-low-power devices. Embedded ML is a new field in which AI-based sensor data analytics is carried out near to where the data are collected in real time. The benchmark presented in [115] focuses on a number of use cases that rely on tiny NNs (i.e., models lower than 100 kB) to analyze sensor data such as audio and video to provide intelligence at the edge of the network. The benchmark consists of four ML tasks that include the use of microphone and camera sensors in different embedded devices:Visual Wake Words (**VWW**): classification task for binary images that detects the presence of a person. For instance, an application use case is in-home security monitoring.Image Classification (**IC**): small image classification benchmark with 10 classes, with several use cases in smart video recognition applications.Keyword Spotting (**KWS**): uses a neural network to detect keywords from a spectrogram, with several use cases in consumer end devices, such as virtual assistants.Anomaly Detection (**AD**): uses a neural network to identify anomalies in machine operating sounds, and has several application cases in industrial manufacturing (e.g., predictive maintenance, asset tracking, and monitoring).

This benchmark aims to measure performance for ML in embedded systems, which operate at a microwatt level and include cameras, wearables, smart sensors, and other IoT devices that demand a certain level of intelligence. Thus, the objective of the benchmark is to measure the performance of such constrained systems in order to achieve higher efficiency over time. The results have been reported based on the embedded ML approach and its hardware and software. Table 3 compares the benchmark results for distinct embedded hardware when running a trained model by measuring the processing latency in milliseconds (i.e., how fast systems can process inputs to produce a valid result) and the respective consumed energy in μJ [116].

### 4.3. Edge-AI Computational Cost

Computation needs for AI are growing rapidly. Recent numbers show that large AI training runs are doubling every 3.5 month and, since 2012, the computational needs have increased by more than 300,000 times [117]. In recent years, a lot of effort has been put into increasing AI accuracy and, especially with DL, accuracy has increased at a steady pace. This increase in accuracy has been very important in making AI a reality in real-world applications; however, to run such high accuracy models, more and more computational resources need to be considered. In the short and medium term, AI will face major challenges that put its sustainability and ecological footprint into perspective. Due to the explosion of its use in several application domains, increased pressure on computational resources is already happening, not only to train but also to run these models, which are increasingly more accurate but also, computationally heavier.

Due to this novel and more sustainable practices regarding AI implementation and deployment are yet to come. In [118], Schwartz et al. introduced the concepts of Red and Green AI, as a way to clarify and distinguish the two major currents AI approaches.

Red AI is known for relying on large models and datasets, as its performance is typically evaluated through accuracy, which is usually obtained through the use of massive processing power. In this context, the relation between model performance and model complexity is known to be logarithmic (i.e., an exponentially bigger model is required for a linear improvement in performance [119]). Furthermore, the quantity of training data and the number of tuning experiments, present the same exponential growth [118]. In each of these cases, a small performance improvement comes at an increased computational cost.

Green AI, on the other hand, focuses on achieving results without increasing or, preferably, lowering computational costs. Unlike Red AI, which results in rapidly increasing computing costs and, as a result, a rising carbon footprint, Green AI has the opposite effect [118]. In Green AI, efficiency is usually prioritized over accuracy when evaluating performance. As a result, Green AI focuses on model efficiency, which includes the amount of effort necessary to create a given result using AI, the amount of work required to train a model, and, if appropriate, the total of all tuning experiments. Efficiency may be assessed using a variety of metrics, including carbon emissions, power consumption, real-time elapsed time, number of parameters, and so on.

### 4.4. Measuring Edge-AI Energy Consumption and Carbon Footprint

The overall cost of using AI can be obtained by considering the resources involved in all processing stages, which include energy consumption and CO_2_ emissions.

#### 4.4.1. Energy Consumption

In [120], Pinto et al. define energy consumption as an accumulation of power dissipation over time:(1)EnergyConsumption=P×t

Note that Energy Consumption is measured in joules and Power (*P*) is measured in watts. The relationship between these two quantities can be easily interpreted through an example: if a software program takes 5 s to execute and dissipates 5 watts, it consumes 25 joules of energy. In the case of software energy consumption, attention must be paid not only to the software under execution, but also to the hardware that executes the software, the environmental context of execution, and its duration.

#### 4.4.2. CO_2_ Emissions

In [121], Strubell et al. presented a study that focused on the estimation of the financial and environmental cost of training a variety of recently successful NN models. To estimate CO_2_ emissions (CO_2_e), they proposed a simple method based on the multiplication of the energy consumption with the average produced CO_2_. After measuring the CO_2_e for several models using different hardware, they concluded that the CO_2_ required for training one model can range from 12 kg up to 284 t. Note that this CO_2_e footprint is highly significant when compared with the world average CO_2_ emissions per capita, whose estimate was 4.56 t in 2016 [122]. Moreover, they evaluated the cost of training these models in the cloud, which raised from USD 41 up to USD 3,201,722, respectively.

### 4.5. Measuring Edge-AI Performance

Although this article focuses on Edge-AI sustainability, there are other factors that should be considered during the evaluation of the performance of an Edge-AI system. Specifically, four main metrics are often used for the performance evaluation of AI algorithms [123]: accuracy, memory bandwidth, energy efficiency, and execution time.

#### 4.5.1. Accuracy

Classification accuracy is the simplest performance metric and is commonly used with balanced datasets (i.e., the number of samples per class is balanced). Accuracy is defined as the number of correct predictions, divided by the total number of predictions, and is implemented by comparing the annotated ground truth data with the predicted results:(2)$$\mathrm{Accuracy}=\frac{tp+tn}{tp+tn+fp+fn}$$
where tp represents the true positives, tn the true negatives, fp are the false positives, and fn the false negatives. Note that, if unbalanced data are considered (i.e., the number of samples per class is not balanced), a new accuracy metric, known as balanced accuracy, should be computed. The balanced accuracy is computed by normalizing tp and tn by the number of positive and negative samples, respectively, then perform their sum, and divide by two, as indicated in Equation (Equation 3):(3)$$\mathrm{Balanced}\;\mathrm{accuracy}=\frac{TP+TN}{2}$$
where TP represents the normalized true positives and TN the normalized true negatives; however, a fair performance evaluation between algorithms should not only rely on the accuracy, as Red AI tends to favor.

#### 4.5.2. Memory Bandwidth

In [124], Jouppi et al. compare the performance of several processors used by Google cloud-based systems on inference tasks when running various types of NNs. The analysis uses a roofline model, where the performance of the algorithms is plotted based on the computational performance (operations per second) versus the operational intensity (number of operations per byte of data). Typically, in cloud-based architectures, the overall performance is limited by the memory bandwidth, and as the operational intensity tends to increase, the performance is limited by the computational capacity of the computer system architecture. Recent hardware architectures, notably SoC architectures, are focused on increasing the memory bandwidth to address the continuously growing demand of AI [98].

#### 4.5.3. Energy Efficiency

A simple metric that can be used to measure the software energy efficiency is presented in [123] and is shown in Equation (Equation 4). In Edge-AI, the useful work performed can be defined as the number of model inferences. As a result, Energy Efficiency can be measured as the number of inferences per Joule.
(4)$$\mathrm{Energy}\;\mathrm{Efficiency}=\frac{\mathrm{Useful}\;\mathrm{Work}\;\mathrm{Performed}}{\mathrm{Energy}\;\mathrm{Consumption}}=\frac{\mathrm{Number}\;\mathrm{of}\;\mathrm{Inferences}}{\mathrm{Energy}\;\mathrm{Consumption}}$$

#### 4.5.4. Execution Time

This metric represents the execution time of a specific task in the ML process to obtain a valid result, which may include, model training or model inference [123], and are measured in seconds, being typically referred as the “training time” and “inference time”, respectively.

## 5. Cross-Analysis of G-IoT and Edge-AI: Key Findings

Although Edge-AI G-IoT system deployment in real-world applications has already started, the research and development are still undergoing, and some issues compromise its wider acceptance, of which we highlight: trustworthiness (e.g., algorithm transparency, traceability, privacy, and data integrity); capacity (e.g., communications bandwidth and coverage, hardware constraints such as power and computational power, security in edge distributed architectures); heterogeneity (e.g., dealing with distinct data sources and formats as well as adapting with a variety of operational, technical, and human requirements); and scale (e.g., inadequate volume of publicly available data, high-quality data required to effectively simulate the physical world’s complexity). In addition, the cross-analysis of the G-IoT and Edge-AI literature allows for obtaining the following key findings that can be useful for future developers and researchers:Communications between G-IoT nodes and Edge-AI devices are essential, so developers should consider the challenges related to the use of energy efficient transceivers and fast-response architectures. Thus, researchers need to contemplate aspects such as the use of low-power communications technologies (e.g., ZigBee, LoRa, UWB, and Wi-Fi Hallow), the management of the RF spectrum or the design of distributed AI training, learning algorithms, and architectures that achieve low-latency inference (either distributed or decentralized [107]).Although the most straightforward way to implement Edge-AI systems is to deploy the entire model on edge devices, which eliminates the need for any communications overhead, when the model size is large or the computational requirements are very high, this approach is unfeasible and it is necessary to include additional techniques that involve the cooperation among nodes to accomplish the different AI training and inference tasks (e.g., federated learning techniques [107]). Such techniques should minimize the network traffic load and communications overhead in resource-constrained devices.Edge-AI G-IoT systems should consider that the different nodes of the architecture (e.g., mist nodes, edge computing devices, and cloudlets) have heterogeneous capabilities in terms of communications, computation, storage, and power; therefore, the tasks to be performed should be distributed in a smart way among the available devices according to their capabilities.Besides heterogeneity, developers should take into account that G-IoT node hardware constrains the performance of the developed Edge-AI systems. Such hardware must be far more powerful than traditional IoT nodes and provide a suitable trade-off between performance and power consumption. In addition, such hardware should be customized to the selected Edge-AI G-IoT architecture and application.Currently, most G-IoT systems rely on traditional cloud computing architectures, which do not meet some of the needs of Edge-AI G-IoT applications in terms of high availability, low latency, high network bandwidth, and low power consumption. Moreover, current cloud-based approaches may be compromised by cyberattacks; therefore, new architectures such as the ones based on fog, mist, and edge computing should be considered to increase the robustness against cyberattacks and to allow for choosing which AI tasks to offload to the cloud, if any, while reducing network resource consumption.Green power sources and energy-harvesting capabilities for Edge-AI G-IoT systems still need to be studied further. Although batteries are typically used to meet power requirements, future developers should analyze the use of renewable power sources or energy-harvesting mechanisms to minimize energy consumption. In addition, the use of decentralized green smart grids for Edge-AI G-IoT architectures can be considered.High-security mechanisms are usually not efficient in terms of energy consumption, so it is important to analyze their performance and carry out practical energy measurements for the developed Edge-AI G-IoT systems.Developers should consider using energy efficiency metrics for the developed AI solutions. For instance, in [123] the authors propose four key indicators for an objective assessment of AI models (i.e., accuracy, memory bandwidth, energy efficiency, and execution time). The trade-off between such metrics will depend on the environment where the model will be employed (e.g., "increased safety" scenarios impose low execution time).

## 6. Application Case: Developing a Smart Workshop

### 6.1. Workshop Characterization and Edge-AI System Main Goals

To illustrate the concepts described in the previous sections, it was selected a practical Industry 5.0 use case in a real-world scenario. Specifically, the selected Industry 5.0 scenario consists in an industrial workshop that looks for improving operator safety through IIoT sensors/actuators and Edge-AI. The chosen scenario is based on the previous work of the authors [125,126,127], which participated in a Joint Research Unit together with one of the largest shipbuilders in the world (Navantia). The specific scenario is the pipe workshop that such a company owns in its shipyard in Ferrol (Spain). The workshop manufactures pipes as follows:First, raw pipes are stored in the Reception Area (shown in Figure 6a). Thus, they are collected by the workers as they are needed. If the pipes are delivered with dirt or grease, then, before being stored in the Reception Area, they are cleaned in the Cleaning Area (in Figure 6b). Operators need to keep away from the Cleaning Area unless authorized because of the presence of dangerous chemical products (e.g., chloridric acid, caustic soda) and water that is pressurized and hot.Second, every pipe is first cut in the Cutting Area according to the required dimensions. Really powerful (and dangerous) mechanical and plasma saws (shown in Figure 7a,b) are used in the Cutting Area. It is important to note that pipes are moved from the Reception Area to the Cutting Area (or from one area to any other area) by stacking them on pallets, which are carried by big gantries installed in the ceiling of the workshop (several pallets can be seen on the foreground of Figure 7b).Third, pipes are bent in the Bending Area. There are three large bending machines in such an area. Operators need to always keep a safe distance and safety glasses when operating a bending machine.Fourth, pipes are cleaned and moved to the Manufacturing Area, where accessories are added. For instance, operators may need to weld a valve to a pipe. Welding requires taking specific safety measures and only the authorized operators can access the welding area when someone is working.Finally, pipes are stacked into pallets, packed, and then stored in two different areas of the workshop (shown in Figure 8a,b).

Figure 9 depicts the main areas of the workshop floor map and shows the position of the IIoT cameras that monitor the presence of the workers. In addition, the dashed semicircles indicate the estimation of the field of view of such cameras. Specifically, Figure 9 shows 18 distinct areas of the factory floor that are equipped with cameras for continuous monitoring (24 h a day, 7 days a week) of a complete manufacturing process. Note that, in this specific application case, images should be neither transmitted nor recorded in the cloud, not only due to bandwidth and connectivity limitations, but also due to the impositions of the General Regulation on Data Protection (GDPR) in force.

The objective of the proposed solution is to harness “visual wake words” in order to detect the presence of the workers with the help of cameras and then lock or unlock the deployed industrial devices and machinery, and automate the available security mechanisms. For instance, industrial robot arms or cutting machines can harm a worker during their operation when safety distance is not respected. Thus, the system takes advantage of the proposed mist AI-enabled architecture (described next in Section 6.2) to achieve two specific application goals:**Increased Safety**: automatically detect humans in the proximity of machinery that is operating. After detection, a sound warning should be physically generated in the surrounding zone. After triggering the sound warning, if the detection persists and the estimated distance between the operating machine and the human does not increase, a shutdown command should be sent to the operating machine.**Operation Tracking**: automatically detect and track human operators and moving machinery. The tracking information is then used for the continuous improvement of manufacturing processes.

Besides the mentioned goals, it is important to note that the proposed system impacts different circular economy aspects:Smarter use of resources: the detection of the presence of operators allows for determining when machinery should be working and when it should be shut down.Reduction of total annual greenhouse gas emissions: the smarter use of resources decreases energy consumption and, as a consequence, carbon footprint.Enhanced process safety: human proximity detection allows for protecting against possible incidents or accidents with the deployed industrial devices and machinery.

### 6.2. System Architecture

The architecture proposed for the application case is shown in Figure 10. As it can be observed, there are two main layers:Mist Computing Layer: it is composed of AI-enabled IIoT nodes that run AI algorithms locally. Thus, after the AI training stage, nodes avoid exchanging image data through the network with edge computing devices or with the cloud, benefiting from:-Lower latency. Since most of the processing is carried out locally, the mist computing device can respond faster.-Communications problems in complex environments can be decreased. Local processing avoids continuous communications with local edge devices or remote clouds. Thus, potential communications problems are reduced, which is really important in industrial scenarios that require wireless communications [126].-Fewer privacy issues. Camera images do not need to be sent to other devices through the network, so potential attacks to such devices or man-in-the-middle attacks can be prevented and thus avoid image leakages.-Improved local communications with other nodes. Mist devices can implement additional logic to communicate directly with other mist devices and machines, so responses and data exchanges are faster, and less traffic is generated due to not needing to make use of intermediate devices such as edge computing servers or the cloud.Despite the benefits of using mist AI-enabled nodes, it is important to note that IIoT nodes, since they integrate cameras/sensors and the control hardware, are more expensive and complex (i.e., there are more hardware parts that can fail).Cloud: it behaves like in the edge computing based architecture. As a consequence, it deals with the requests of the mist devices that cannot be handled locally.
Figure 10Mist-computing-based communications architecture.
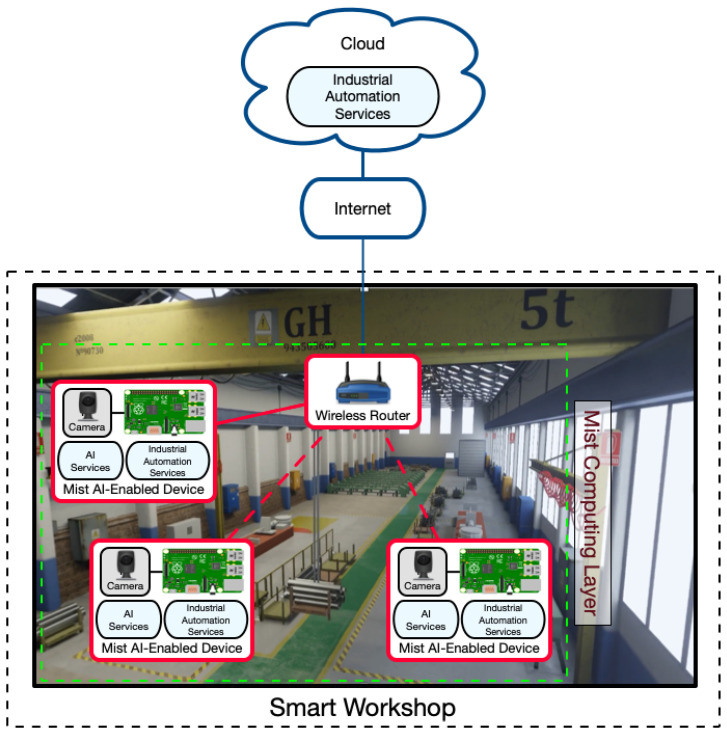


### 6.3. Energy Consumption of the Mist AI-Enabled Model

In this application case, latency is a critical factor, and a low fault-tolerance policy needs to be implemented. To achieve the “Increase Safety” goal, the use of object detection models with low inference latencies is mandatory. In this case, the human movement dynamics are typically low, since, running on the factory floor is typically not allowed. Moreover, with respect to the “Operations Tracking” goal, the inference latency is not critical, since it does not affect the obtained results, due to the deterministic nature of the inference latency.

To estimate the energy cost of the overall system, it was considered the data presented in Table 3 for an STM32 Nucleo-L4R5ZI processor running TensorFlow Lite with a Mobinet-V1 model (Task #1-Visual Wake Words) to simulate the “Increase Safety” task and a Resnet-V1 model (Task #2-Image Classification) for simulating the “Operations Tracking” task. The former is a classification task for binary images that detect the presence of a person with an inference latency of 603.14 ms and energy consumption of 24,320.84 μJ per inference (1 joule = 2.77777778 × 10${}^{-7}$ kWh). The latter is an image classification benchmark with 10 classes for smart video recognition applications with an inference latency of 704.23 ms and energy consumption of 29,207.84 μJ per inference. At this stage, it is important to notice that only inference is being considered, since no information is available regarding the training stage, namely the consumed energy.

First, the number of inferences can be estimated for a year and one camera, and then the overall power consumption can be extrapolated to all cameras, based on the previous assumptions:(5)$$N_{VWW}=\frac{365\times 24\times 3600s}{603.14\mathrm{ms}}=52,286,368\;\mathrm{inferences}/\mathrm{year}$$
(6)$$E_{VWW}=N_{VWW}\times 24,320.84\;\mu J=12,716,483.9\;J=0.353\;\mathrm{kWh}/\mathrm{device}$$
(7)$$N_{IC}=\frac{365\times 24\times 3600\;s}{704.23\;\mathrm{ms}}=44,780,824\;\mathrm{inferences}/\mathrm{year}$$
(8)$$E_{IC}=N_{IC}\times 29,207.84\;\mu J=1,307,951.2\;J=0.363\;\mathrm{kWh}/\mathrm{device}$$
where $N_{x}$ represents the number of inferences per year for model *x* (VWW or IC) and $E_{x}$ represents the total equivalent energy consumed in one year per device. In this particular case, the energy refers only to the one consumed by the inference task. Given that, in this study, we are only focused on the additional power consumption of the inference stage, the power consumed by all functional hardware blocks has not been included.

Equation (Equation 6) indicates that each camera, when running the VWW model, consumes approximately 0.353 kWh in a year. When running the IC model for the same period (Equation (Equation 8)), each camera consumes approximately 0.363 kWh; therefore, by extrapolating for the 18 cameras, we achieve a total consumption (in one year) of 6.354 kWh and 6.534 kWh, for the VWW and IC models, respectively. This power consumption is on the Green-AI magnitude scale, and the yearly inference cost of all the 18 cameras can easily be maintained by a conventional renewable energy source, such as a photovoltaic panel.

### 6.4. Carbon Footprint

Carbon footprint can be estimated by using the formula in Equation (Equation 9) [128]:(9)$$CO_{2}e\left(g\right)=E_{x}\left(KWh\right)\times I_{N}\left(g/KWh\right)$$
where $CO_{2}e$ is the number of grams of emitted CO_2_, $E_{x}$ (*x* equal to VWW or IC) is the consumed energy (in KWh) and $I_{N}$ is the carbon intensity (in grams of emitted CO_2_ per KWh). This latter parameter can be obtained through the data published publicly by many countries or by organizations such as the European Union, but it is easier to obtain it from Electricity Maps [129], an open-source project that collects such data automatically and plots them through a user-friendly interface. Such a website also indicates the energy sources used by each country (an example of such sources for France, Portugal, Spain, California, and the province of Alberta is shown in Figure 11). The data were obtained for 25 July 2021 and, as it can be observed, energy sources differ significantly from one country to another:France (data source: Réseau de Transport d’Electricité (RTE)): it has almost got rid of CO_2_-intensive energy sources thanks to generating most of its electricity through nuclear power. Nonetheless, on 25 July 2021, when the data in Figure 11 were collected, only roughly 31% of France’s energy came from renewable sources.Portugal (data source: European Network of Transmission System Operators for Electricity (ENTSOE)): its most relevant energy source is natural gas, but, when the data were gathered, approximately 43% of its energy came from renewable sources and none from nuclear power.Spain (data source: ENTSOE): like Portugal, it has a dependency on natural gas, but, thanks to a powerful solar energy sector, it generates roughly 53% of its energy from renewable sources. In addition, almost 24% of the Spanish energy comes from nuclear power, so a total of 77% of the energy is generated from low-carbon technologies.California (data source: California Independent System Operator (CAISO)): in spite of being a state of the U.S., it was selected due to its crucial role in IT and cloud-based services. Nearly 42% of its energy on 25 July 2021 was generated through low-carbon technologies, but almost 58% came from natural gas.Alberta (data source: Alberta Electric System Operator (AESO)): it was included as an example of a rich area with a key role in the oil and natural gas production in North America. As it can be observed in Figure 11, most of its energy (almost 84%) is generated by natural gas and coal, which results in the generation of a large amount of CO_2_ emissions.
Figure 11Energy sources for France, Portugal, Spain, California, and Alberta (25 July 2021).
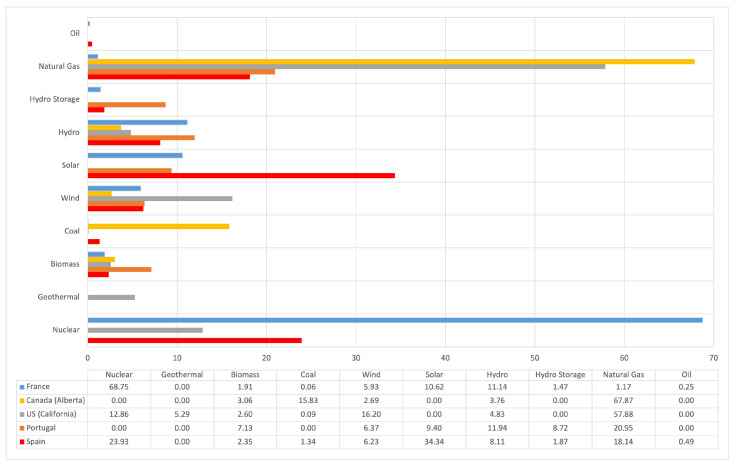


Figure 12 shows the estimated CO_2_ emissions for the energy consumption estimated in the previous section. As it can be easily guessed, emissions increase with the number of deployed mist AI-enabled devices; however, such growth changes dramatically from one country to another depending on the energy source: while near-zero emission countries like France are barely impacted by the increase in the number of deployed devices, a province like Alberta emits more than 17 times more CO_2_ for 1000 deployed devices.

It is also possible to obtain the monetary cost of running the mist AI-enabled devices (as an example, the average prices for April 2021 for each territory were considered), which is depicted in Figure 13. As it can be seen in the figure, the cost of running the system in Alberta would be cheaper but will result in more CO_2_ emissions. In contrast, the countries with the largest shares of renewable energy sources (Spain and Portugal) are the ones with the most expensive electricity. Nonetheless, please note that such a link between the use of renewable energies and cost is impacted by other external factors (e.g., taxes, environmental policy, and energy trading).

## 7. Future Challenges of Edge-AI G-IoT Systems

Despite the promising foreseen future of Edge-AI G-IoT systems, it is possible to highlight some open challenges that must be faced by future researchers:Additional mechanisms are needed to offer protection against network, physical, software, and encryption attacks. In addition, it is critical to have protection against adversarial attacks during on-device learning [130].Future communications networks. 5G/6G are intended to deliver low-latency communications and large capacity; therefore, moving the processing tasks to the network edge will demand higher edge computing power, which puts G-IoT and Edge-AI convergence as fundamental technology enablers for the next 6G mobile infrastructure. Moreover, the rapid proliferation of new products and devices and their native connectivity (at a global level) will force the convergence of not only G-IoT and Edge-AI, but also 5G/6G communication technologies, the latter being a fundamental prerequisite for future deployments. Indeed, future communications services should also provide better dependability and increased flexibility to effectively cope with a continuously changing environment.Edge-AI G-IoT Infrastructure. The IoT market is currently fragmented, so it is necessary to provide a comprehensive standardized framework that can handle all the requirements of Edge-AI G-IoT systems.Decentralized storage. Cloud architectures store data in remote data centers and digital infrastructures that require substantial levels of energy. Luckily, recent architectures for Edge-AI G-IoT systems are able to decentralize data storage to prevent cyberattacks and avoid high operating costs; however, to achieve energy optimizations for such decentralized architectures, sophisticated P2P protocols are needed.G-IoT supply chain visibility and transparency. To increase the adoption of the DCE and limit the environmental impact of a huge number of connected devices, further integration of value chains and digital enabling technologies (e.g., functional electronics, UAVs, blockchain) is needed. End-to-end trustworthy G-IoT supply chains that produce, utilize, and recycle efficiently are required.Development of Edge-AI G-IoT applications for Industry 5.0. The applications to be developed should be first analyzed in terms of its critical requirements (e.g., latency, fault tolerance) together with the appropriate communications architecture, while considering its alignment with social fairness, sustainability, and environmental impact. In addition, hardware should be customized to the selected Edge-AI G-IoT architecture and the specific application.Complete energy consumption assessment. For the sake of fairness, researchers should consider the energy consumption of all the components and subsystems involved in an Edge-AI G-IoT system (e.g., communications hardware, remote processing, communications protocols, communications infrastructure, G-IoT nodes, and data storage), which may be difficult in some practical scenarios and when using global networks.Digital circular life cycle of Edge-AI G-IoT systems. In order to assess the impact of circular economy based applications, all the different stages of the digital circular life cycle (i.e., design, development, prototyping, testing, manufacturing, distribution, operation, maintenance, and recycling stages) should be contemplated.CO_2_ emission minimization for large-scale deployments. Future developers will need to consider that CO_2_ emissions increase with the number of deployed Edge-AI IoT devices. In addition, such growth changes dramatically from one country to another depending on the available energy source.Corporate governance, corporate strategy, and culture. Organization willingness to explore new business strategies and long-term investments will be critical in the adoption of Edge-AI G-IoT systems, as a collaborative approach is required to involve all the stakeholders and establish new ways for creating value while reducing the carbon footprint. New business models will emerge (e.g., Edge-AI as a service, such as NVIDIA Clara [131]).

## 8. Conclusions

This article reviewed the essential concepts related to the development of Edge-AI G-IoT systems and their carbon footprint. In particular, the most relevant Edge-AI G-IoT communications architectures were analyzed together with their main subsystems. In addition, the most recent trends on the convergence of AI and edge computing were analyzed and a cross-analysis on the fusion of Edge-AI and G-IoT was provided. Furthermore, an Industry 5.0 application case was described and evaluated in order to illustrate the theoretical concepts described throughout the article. The obtained results show how CO_2_ emissions increase depending on the number of deployed Edge-AI G-IoT devices and on how greener is the energy generated by a country. Finally, the main open challenges for the development of the next generation of Edge-AI G-IoT systems were enumerated to guide future researchers.

## Figures and Tables

**Figure 1 sensors-21-05745-f001:**
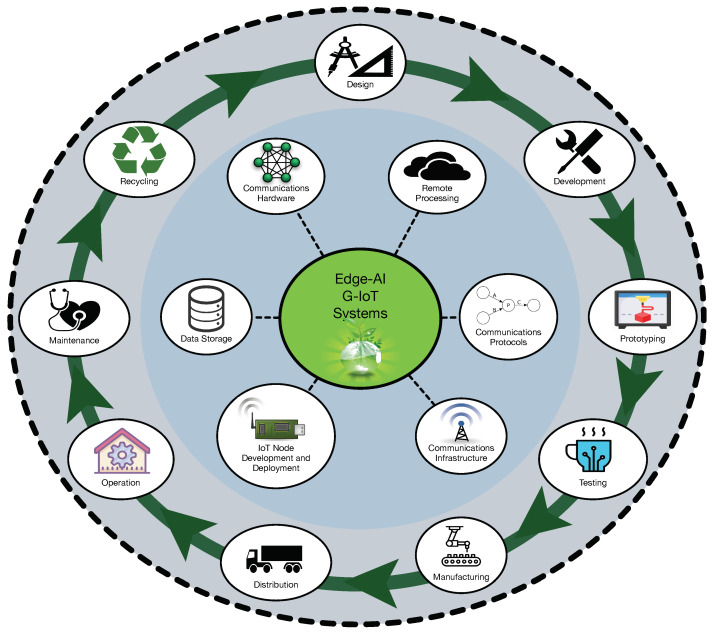
Edge-AI G-IoT main areas and their digital circular life cycle.

**Figure 2 sensors-21-05745-f002:**
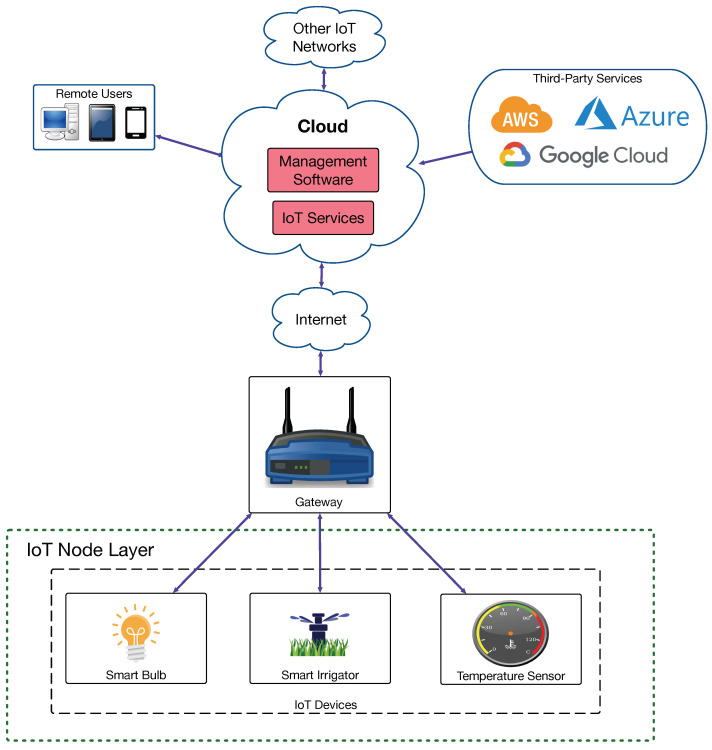
Cloud-based IoT architecture.

**Figure 3 sensors-21-05745-f003:**
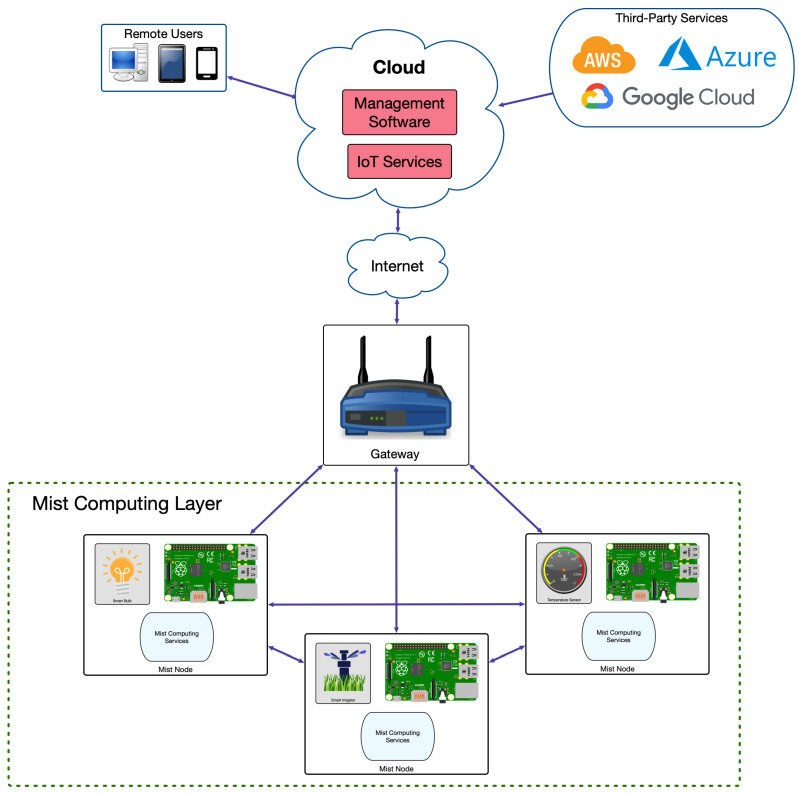
Example of mist computing architecture.

**Figure 4 sensors-21-05745-f004:**
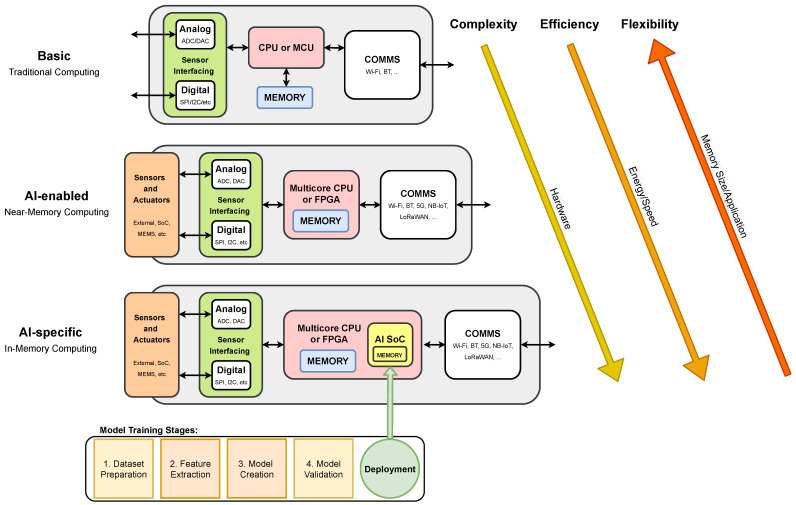
Basic, AI-enabled and AI-specific IoT device architectures.

**Figure 5 sensors-21-05745-f005:**
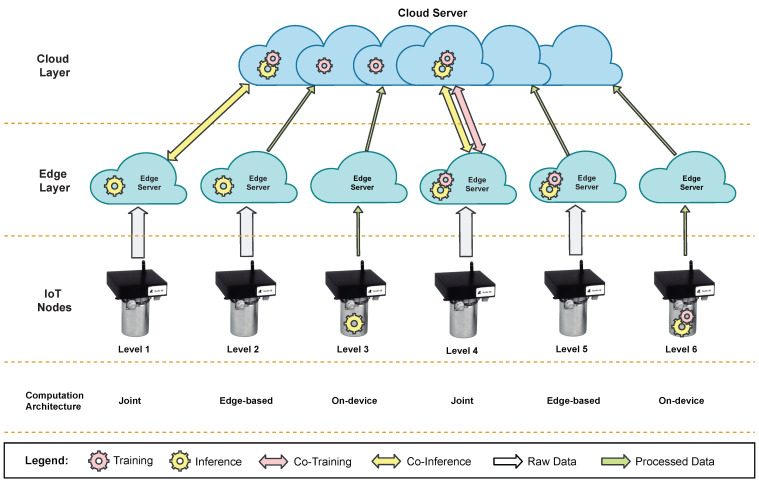
Edge-AI Levels and model inference computation architectures: on-device, edge-based, and joint.

**Figure 6 sensors-21-05745-f006:**
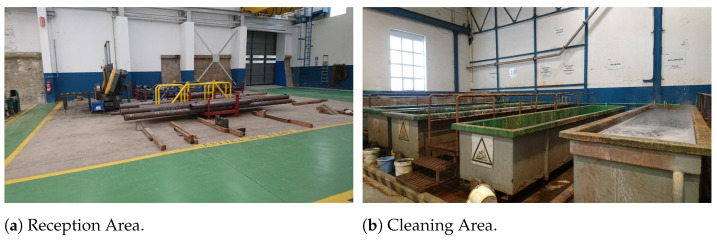
Relevant areas of the workshop.

**Figure 7 sensors-21-05745-f007:**
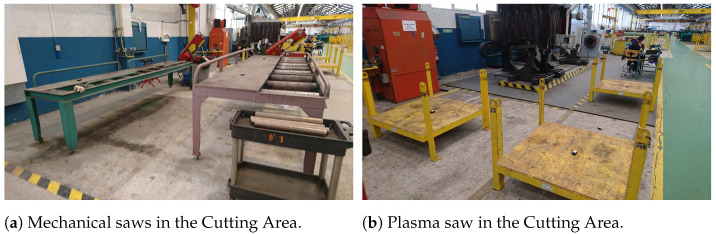
Saws of the Cutting Area.

**Figure 8 sensors-21-05745-f008:**
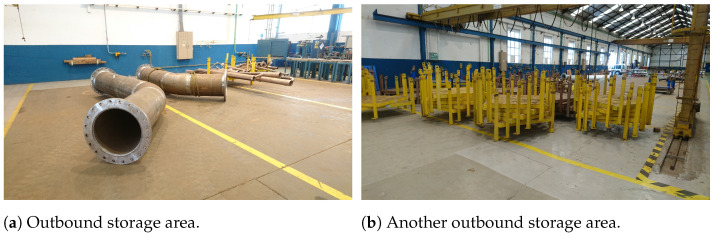
Main storage areas.

**Figure 9 sensors-21-05745-f009:**
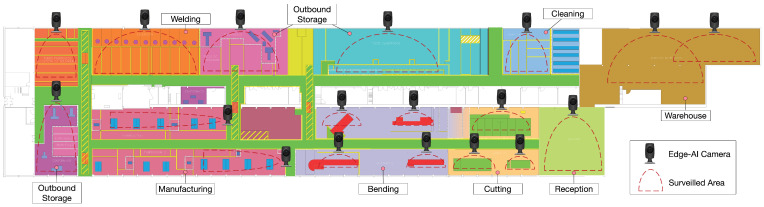
Floor map of the smart workshop.

**Figure 12 sensors-21-05745-f012:**
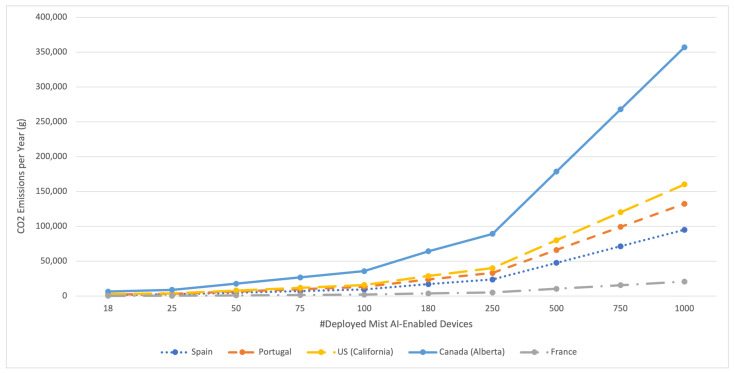
Estimated CO_2_ emissions for different number of deployed devices for different countries.

**Figure 13 sensors-21-05745-f013:**
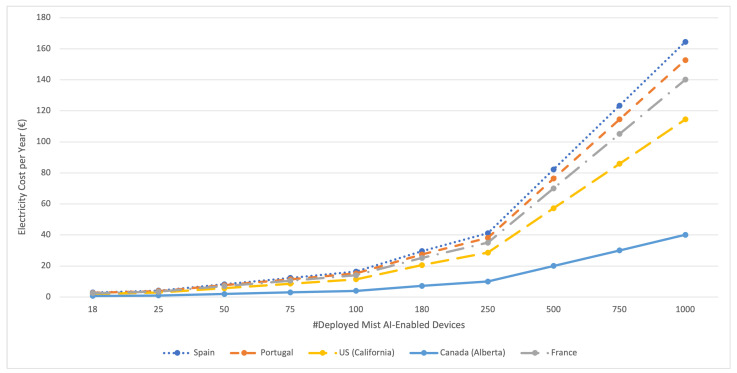
Electricity cost for different number of deployed devices and for different countries.

**Table 1 sensors-21-05745-t001:** Main characteristics of the most relevant communications technologies for G-IoT nodes.

Technology	Power Consumption	Frequency Band	Maximum Range	Data Rate	Main Features	Popular Applications
NFC	Tags require no batteries, no power	13.56 MHz	<20 cm	424 kbit/s	Low cost	Ticketing and payments
Bluetooth 5 LE	1–20 mW, Low power and rechargeable (days to weeks)	2.4 GHz	<400 m	1360 kbit/s	Trade-off among different PHY modes	Beacons, wireless headsets
EnOcean	Very low consumption or battery-less thanks to using energy harvesting	868–915 MHz	300 m	120 kbit/s	Up to $2^{32}$ nodes	Energy harvesting building automation applications
HF RFID	Tags require no batteries	3–30 MHz (13.56 MHz)	a few meters	<640 kbit/s	NLOS, low cost	Smart Industry, payments, asset tracking
LF RFID	Tags require no batteries	30–300 KHz (125 KHz)	<10 cm	<640 kbit/s	NLOS, durability, low cost	Smart Industry and security access
UHF RFID	Batteries last from days to years	30 MHz–3 GHz	tens of meters	<640 kbit/s	NLOS, durability, low cost	Smart Industry, asset tracking and toll payment
UWB/IEEE 802.15.3a	Low power, rechargeable (hours to days)	3.1 to 10.6 GHz	<10 m	>110 Mbit/s	Low interference	Fine location, short-distance streaming
Wi-Fi (IEEE 802.11b/g/n/ac)	High power consumption, rechargeable (hours)	2.4–5 GHz	<150 m	up to 433 Mbit/s (one stream)	High-speed, ubiquity, easy to deploy and access	Wireless LAN connectivity, Internet access
Wi-Fi HaLow/IEEE 802.11ah	Power consumption of 1 mW	868–915 MHz	<1 km	100 Kbit/s per channel	Low power, different QoS levels (8192 stations per AP)	IoT applications
ZigBee	Very low power consumption, 100–500 μW, batteries last months to years	868–915 MHz, 2.4 GHz	<100 m	Up to 250 kbit/s	Up to 65,536 nodes	Smart Home and industrial applications
LoRa	Long battery life, it lasts >10 years	2.4 GHz	kilometers	0.25−50 kbit/s	High range, resistant to interference	Smart cities, M2M applications
SigFox	Battery lasts 10 years sending 1 message, <10 years sending 6 messages	868–902 MHz	50 km	100 kbit/s	Global cellular network	M2M applications

**Table 2 sensors-21-05745-t002:** AI-enabled IoT hardware compatible with TensorFlow Lite.

Board	Processor	Power	Connectivity	Architecture Type	Cryptographic Engine	Cost
Arduino Nano 33 BLE Sense [101]	ARM Cortex-M0 32-bit@64 MHz	52 μA/MHz	BLE	AI-enabled	Yes	€27
SparkFun Edge [102]	ARM Cortex-M4F 32-bit@48/96 MHz	6 μA/MHz	BLE 5	AI-enabled	Yes	€15
Adafruit EdgeBadge [103]	ATSAMD51J19A 32-bit@120 MHz	65 μA/MHz	BLE/WiFi	AI-enabled	Yes	€35
ESP32-DevKitC [104]	Xtensa dual-core 32-bit@160/240 MHz	2 mA/MHz	BLE/WiFi	AI-enabled	Yes	€10
ESPEYE-DevKit [105]	Xtensa dual-core 32-bit@160/240 MHz	2 mA/MHz	BLE/WiFi	AI-enabled	Yes	€50
STM32 Nucleo-144 [106]	ARM Cortex-M4 Nucleo-L4R5ZI 32-bit@160/120 MHz	43 μA/MHz	Ethernet	AI-enabled	No	€100

**Table 3 sensors-21-05745-t003:** MLPerf™ Tiny Inference v0.5 benchmark results. Data from [115].

ID	Submitter	Device	Processor	Software	Results								
					Task	#1 - VWW		#2 - IC		#3 - KS		#4 - AD	
					Data	Visual Wake Words Dataset		CIFAR-10		Google Speech Commands		ToyADMOS (ToyCar)	
					Model	MobileNetV1 (0.25x)		ResNet-V1		DSCNN		FC AutoEncoder	
					Accuracy	80% (Top 1)		85% (Top 1)		90% (Top 1)		0.85 (AUC)	
					Units	Latency (ms)	Energy (uJ)	Latency (ms)	Energy (uJ)	Latency (ms)	Energy (uJ)	Latency (ms)	Energy (uJ)
1l0.5-464	Harvard (Reference)	Nucleo-L4R5ZI	Arm Cortex M4 w/ FPU	Tensorflow Lite for Microcontrollers		603.14	24,320.84	704.23	29,207.01	181.92	7373.70	10.40	416.31
0.5-465	Peng Cheng Laboratory	PCL Scepu02	RV32IMAC with FPU	TensorFlowLite for Microcontrollers 2.3.1 (modified)		846.74	-	1239.16	-	325.63	-	13.65	-
1l0.5-466	Latent AI	RPi 4	Broadcom BCM2711	LEIP Framework		3.75	-	1.31	-	0.39	-	0.17	-
0.5-467	Latent AI	RPi 4	Broadcom BCM2711	LEIP Framework		2.60	-	1.07	-	0.42	-	0.19	-

## Data Availability

Not applicable.

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
