# Peer review of "Green IoT and Edge AI as Key Technological Enablers for a Sustainable Digital Transition towards a Smart Circular Economy: An Industry 5.0 Use Case"

_sensors, 2021, doi:10.3390/s21175745_

Round 1

Reviewer 1 Report

The authors made an excellent compilation of existing work in a relevant and timely subject. The paper is very well written, however too lengthy. The authors presented a very good summarization of emerging trends and research priorities of G-IoT and Edge-AI, bringing many valuable insights. It has a clear contribution as a secondary study. The energy consumption analysis presented in the paper is also very useful and the presented use case properly illustrate the concepts included in the paper. As a weakness of the paper, it falls short of providing in-depth, cross-analysis on the addressed subject. The authors could decrease a little bit the descriptive content and improve the analytics content of the paper. As a survey type paper, the main contribution should be revealing new facts beyond just gathering findings from the original studies described.

Author Response

The authors would like to thank the reviewer for his/her valuable comments, which have certainly helped us to improve the manuscript. Please find attached our detailed responses to the comments. In order to ease the labor of the reviewers we have colored in red the differences with the previous version of the article.

Reviewer 2 Report

While the paper is very interesting considering a good example of AI in the edge. The introduction and first section in circular economy are not supported enough of the rest of the paper.

The  part of mist computing can be referenced in well know publications in the area such us : 10.1109/MCOM.2017.1600730 or 10.3390/s17091978.

Additionally we propose to the authors to see the paradigm of  hardware acceleration platform for AI-based inference at the edge that can show in a clear manner the added value in energy reduce.

The paper also needs to restructured for making the life of the reader easier by explaining what is the contribution of the paper  to the academic society.

Small grammar and large sentences needs to be checked.

Author Response

(The authors gave the same response as above.)

Round 2

Reviewer 2 Report

The authors addressed all my comments